# A Review on the Role of Non-Coding RNAs in the Pathogenesis of Myasthenia Gravis

**DOI:** 10.3390/ijms222312964

**Published:** 2021-11-30

**Authors:** Soudeh Ghafouri-Fard, Tahereh Azimi, Bashdar Mahmud Hussen, Mohammad Taheri, Reza Jalili Khoshnoud

**Affiliations:** 1Department of Medical Genetics, School of Medicine, Shahid Beheshti University of Medical Sciences, Tehran 19835-35511, Iran; s.ghafourifard@sbmu.ac.ir; 2Phytochemistry Research Center, Shahid Beheshti University of Medical Sciences, Tehran 19835-35511, Iran; Azimi@sbmu.ac.ir; 3Department of Pharmacognosy, College of Pharmacy, Hawler Medical University, Erbil 44001, Iraq; Bashdar.Hussen@hmu.edu.krd; 4Institute of Human Genetics, Jena University Hospital, 07747 Jena, Germany; 5Functional Neurosurgery Research Center, Shahid Beheshti University of Medical Sciences, Tehran 19835-35511, Iran

**Keywords:** myasthenia gravis, lncRNA, miRNA, expression, biomarkers

## Abstract

Myasthenia gravis (MG) is an autoimmune condition related to autoantibodies against certain proteins in the postsynaptic membranes in the neuromuscular junction. This disorder has a multifactorial inheritance. The connection between environmental and genetic factors can be established by epigenetic factors, such as microRNAs (miRNAs) and long non-coding RNAs (lncRNAs). XLOC_003810, SNHG16, IFNG-AS1, and MALAT-1 are among the lncRNAs with a possible role in the pathoetiology of MG. Moreover, miR-150-5p, miR-155, miR-146a-5p, miR-20b, miR-21-5p, miR-126, let-7a-5p, and let-7f-5p are among miRNAs whose roles in the pathogenesis of MG has been assessed. In the current review, we summarize the impact of miRNAs and lncRNAs in the development or progression of MG.

## 1. Introduction

Myasthenia gravis (MG) is an autoimmune condition caused by the presence of autoantibodies against certain proteins in the postsynaptic membranes in the neuromuscular junction [1]. The main detected autoantibodies are targeted against the muscle acetylcholine receptor (AChR) [2]. Other targets of autoantibodies are MuSK [3] and LRP4 [4]. The observed 35% concordance rate in MG occurrence among monozygotic twins shows the impact of environmental factors in the pathoetiology of this disorder [5]. From a clinical point of view, MG is characterized by fluctuating fatigability and weakness in a number of muscles, such as ocular, bulbar, and limb muscles. MG subtypes include ocular myasthenia and generalized myasthenia, which can have mild, moderate, or severe presentations [6]. Genetic studies have revealed the role of HLA loci as well as a number of other genes such as TNIP1 and PTPN22 in the pathogenesis of MG [5]. Moreover, pathways regulating differentiation of regulatory T cells as well as NF-κB signaling are implicated in this disorder. Significant heterogeneity has been detected in the course of MG in terms of epitopes targeted by autoantibodies, age of disease onset, and thymus histopathology [7]. While early onset MG cases are predominantly females with hyperplastic thymus histology, late onset cases are mainly males having normal or atrophic thymus [8].

Taken together, MG is a multifactorial disorder with both genetic and environmental etiologies. Meanwhile, the connection between environmental and genetic factors can be established by epigenetic factors [5], such as microRNAs (miRNAs) and long non-coding RNAs (lncRNAs). In the current review, we summarize the impact of miRNAs and lncRNAs in the development or progression of MG.

## 2. LncRNAs and MG

LncRNAs are transcripts with sizes ranging from two hundred nucleotides to thousands of nucleotides. While they share several features with mRNAs, they lack the ability to be served as templates for proteins. Instead, they regulate expression of protein-coding genes at different levels [9]. Dysregulation of lncRNAs have been reported in MG. For instance, experiments in thymus specimens of patients with MG and MG-thymoma (MG-T) have shown upregulation of XLOC_003810 lncRNAs in these patients parallel with increased frequency of CD4+ T cells and release of proinflammatory cytokines in these patients. Functionally, XLOC_003810 upregulation has enhanced frequency of CD4+ T cells, increased synthesis of inflammatory cytokines, and reduced CD4+ PD-1+ T cells and CD14+ PD-L1+ monocytes in the mononuclear cells of thymus. XLOC_003810 silencing has resulted in opposite effects. Cumulatively, XLOC_003810 is an lncRNA that enhances T cells activation and suppresses PD-1/PD-L1 signaling in MG-T patients [10]. Another study has shown higher levels of XLOC_003810 in thymic CD4+ T cells obtained from MG-T patients compared with the control group. Additionally, MG-T thymic CD4+ T cells have exhibited a higher Th17/Treg ratio, higher frequency of Th17 cells and increased expression of Th17-associated proteins, while exhibiting lower amounts of Treg cells and downregulation of Treg-associated proteins. Notably, upregulation of XLOC_003810 has aggravated the imbalance between Th17 and Treg cells in MG-T thymic CD4+ T cells. Thus, XLOC_003810 could influence the balance between Th17 and Treg cells in the context of MG-T [11].

SNHG16 is another lncRNA that partakes in the pathogenesis of MG. This lncRNA acts as a competing endogenous RNA (ceRNA). Expression of SNHG16 has been found to be increased in peripheral blood mononuclear cells (PBMCs) of MG cases compared with controls. Mechanistically, SNHG16 increases expression of IL-10 through acting as a ceRNA for let-7c-5p. Moreover, SNHG16 could inhibit apoptotic processes in Jurkat cells and increase their proliferation through sponging this miRNA [12].

In MG patients, IFNG-AS1 is another dysregulated lncRNA whose expression is associated with the specific quantitative scoring system for MG, i.e., QMG as well as the presence of anti-AchR Ab antibody. Experiments in an animal model of MG have shown that this lncRNA affects proliferation of Th1/Treg cells and regulates expression of Th1/Treg-related transcription factors. Moreover, IFNG-AS1 has been found to decrease expressions of HLA-DRB and HLA-DOB. Moreover, it can affect the expression of CD40L and activity of CD4+ T cells through influencing HLA-DRB1 expression. Thus, IFNG-AS1 has a possible role in regulation of CD4+ T cell-associated immune response in MG [13].

Another study has reported downregulation of MALAT-1 in MG. This lncRNA has been found to compete with MSL2 for binding to miR-338-3p. Thus, MALAT-1/miR-338-3p/MSL2 axis is a new interaction network in MG [14].

A high throughput study in MG has shown upregulation of more than 1500 lncRNAs and downregulation of more than 1000 lncRNAs, parallel with dysregulation of several mRNAs in these patients. Dysregulated genes have been involved in MG-related cellular processes such as nucleic acid transcription factor activity, inflammatory responses, leukocytes activation, and lymphocytes proliferation. Moreover, they have been enriched in pathways related to immune function such as cytokine–cytokine receptor interaction, intestinal immune network for IgA synthesis, NOD-like receptor signaling, and cell adhesion, as well as NF-κB and TNF signaling pathways [15].

Another high throughput study has revealed differential expression of several lncRNAs in MG patients versus MG-T patients—among them has been lncRNA oebiotech_11933, whose dysregulation has been confirmed by real-time PCR. Cell responses to IFN-γ, platelet degranulation, chemokine receptor binding, and cytokine interactions have been identified as processes being involved in the pathoetiology of MG. Transcription factors such as CTCF, TAF1, and MYC as *trans*-regulatory mechanism for modulation of lncRNAs and genes, showing an important transcription factor–lncRNA–target gene network in the context MG [16]. Table 1 summarizes the role of lncRNAs in MG.

## 3. miRNAs and MG

miRNAs are small regulatory molecules acting at post-transcriptional level to finely modulate the expression of genes. miRNAs can participate in the evolution of MG through different mechanisms. A system biology approach in the context of MG has led to construction of transcription factor/miRNA/gene network. Subsequent analyses have resulted in extraction of 5 genes, 3 transcription factors, and 13 miRNAs. Notably, MYC has been identified as the key transcription factor in MG. The identified genes and miRNAs were principally enriched in cancer- and infection-associated pathways. In addition, the composite feed-forward loop motif-specific subnetwork has shown the potential beneficial impact of estradiol, estramustine, raloxifene, and tamoxifen in treatment of MG [20].

An integrative bioinformatics approach and literature search has led to identification of 41 MG-associated signaling pathways and 105 medications that can influence these pathways. Notably, MG-associated miRNAs and drugs can affect key MG-associated pathways, including cytokine–cytokine receptor interaction. This approach has also shown that rituximab, adalimumab, sunitinib, and muromonab might influence MG course, potentiating these drugs as novel treatments for MG [21].

Expression profiling of samples obtained from orbicularis oculi muscle (OOM) and paralyzed extraocular muscle (EOM) of patients with ophthalmoplegic MG has shown similar expression profiles of transcripts among OOM and EOM samples. Ophthalmoplegic MG cases have exhibited downregulation of eight genes in OOM samples compared with controls. The mitochondrial transcription factor TFAM has been among these genes. Notably, numerous miRNAs known to be upregulated in EOM samples have been predicted to affect expression of a number of these genes [22].

Additionally, MG risk pathways such as T cell receptor and Toll-like receptor pathways as well as natural killer cell-mediated cytotoxicity have been shown to be targeted by several miRNAs. Notably, a number of miRSNPs “switches” have been found that affect miRNA regulation in the MG-associated pathways. These miRSNPs can affect gene expressions and pathway activities. An example of these SNPS is rs28457673 (miR-15/16/195/424/497 family), which affects IGF1R expression [23]. Another bioinformatics study has predicted the effects of 18 MG-associated miRNAs on expression of MAPK1, SMAD4, SMAD2, and BCL2 and their subsequent impact on cellular pathways associated with adherens, junctions, apoptosis, or cancer-related features. These important genes negatively regulate T cell differentiation [24].

Another miRNA profiling study has shown differential expression of 41 miRNAs among MG patients who respond to immunosuppressive treatments versus non-responders. Three miRNAs clustered on 14q32.31—namely, miR-323b-3p, miR-409-3p, and miR-485-3p have been validated to be expressed at lower levels in non-responder compared with responders, while miR-181d-5p and miR-340-3p have exhibited the opposite trend. miR-323b-3p, miR-409-3p, and miR-485-3p have been suggested as markers for the prediction of response to immunosuppressive treatment in MG. Five miRNAs have been predicted to be associated with immune functions and drug metabolism. In brief, miR-323b-3p, miR-409-3p, miR-485-3p, miR-181d-5p, and miR-340-3p expression profiles are correlated with therapeutic response in MG patients [25]. Another investigation has reported downregulation of miR-320a in MG patients compared with control subjects, parallel with upregulation of pro-inflammatory cytokines in these patients. MAPK1 has been identified as a direct target of miR-320a. miR-320a downregulation has led to upregulation of pro-inflammatory cytokines via increasing expression of COX-2. This process was regulated by ERK and NF-κB signaling pathways [26]. Table 2 shows the role of miRNAs in MG.

## 4. Discussion

MG is an autoimmune disorder with multifactorial inheritance. Epigenetic factors such as regulatory non-coding RNAs have been found to affect pathogenesis of this disorder, possibly linking between environmental and genetic factors. LncRNAs can affect immune cells phenotypes, modulate balance between Th17 and Treg cells, and regulate expression of proinflammatory cytokines. miRNAs have been more investigated in the context of MG compared with lncRNAs. A number of lncRNAs that have been found to contribute to MG pathogenesis act as ceRNAs for miRNAs, further highlighting the impact of miRNAs in MG. Examples of lncRNA/miRNA/mRNA axes in participating in the pathogenesis of MG are SNHG16/let-7c-5p/IL-10 and MALAT-1/miR-338-3p/MSL2. In addition, high throughput studies have established some transcription factor/lncRNA/target gene networks in the context of MG, adding a layer of complexity in regulation of function of ncRNAs in this context.

Several of the lncRNAs and miRNAs that affect pathogenesis of MG converge on NF-κB and TNF signaling pathways. Moreover, expression profiles of these transcripts have been changed in response to immunomodulatory therapies, particularly corticosteroids.

A number of miRNAs have differential expression between different classes of MG, particularly those with and without thymoma. Moreover, the expression profile of some ncRNAs such as miR-106a-5p, miR-23b, miR-27a-3p, and IFNG-AS1 has been correlated with disease severity and QMG score, suggesting their participation in the pathoetiology of MG.

NcRNAs, particularly miRNAs, have the potential to be used as markers for the predication of response of MG patients to prescribed drugs and for the stratification of patients based on this issue to design patient-specific therapeutic regimens. Moreover, expression of these transcripts might be different during distinct stages of MG, potentiating these transcripts as biomarkers for differentiation of disease status. In spite of extensive research in this field, it is not clear how the expression profile of these transcripts can affect the course of disorder or define the muscle groups that are affected during disease course.

Due to the complex nature of participation of genetic factors in the pathogenesis of MG, high throughput studies are needed to find factors acting at upstream and downstream of lncRNAs and miRNAs. This type of study will reveal novel transcription factor/lncRNA/miRNA/mRNA axes with putative roles in the pathophysiology of MG. These molecular axes represent therapeutic targets for MG. Moreover, system biology approaches have potential for the discovery of novel drugs for the treatment of MG.

The impact of miRNAs on drug response has been suggested through both system biology and experimental studies, potentiating these transcripts as targets for therapeutic interventions. Taken together, miRNAs and lncRNAs partake in the pathoetiology of MG, disease course, and response to immunosuppressive treatments. Thus, these transcripts can be used as markers for prediction of these aspects.

## 5. Conclusions and Future Perspectives

As an autoimmune disorder, MG is associated with dysregulation of miRNAs and lncRNAs. However, the therapeutic implications of these transcripts in MG are not clear. The biomarker role of non-coding RNAs in MG has been investigated. Nonetheless, there is no clear evidence of whether the expression profile of these transcripts can determine disease course or involvement of certain groups of muscles. Future high throughput sequencing experiments should unravel differential expression of ncRNAs during different stages of MG.

## Figures and Tables

**Table 1 ijms-22-12964-t001:** Role of lncRNAs in MG (PBMC: peripheral blood mononuclear cell, qRT-PCR: quantitative reverse transcription PCR, HC: healthy control, NC: negative control (cardiac surgery cases), MG-T: myasthenia gravis-related thymoma, QMGS: Quantitative Myasthenia Gravis Score).

lncRNA	Participants	Source of Materials/Methods	Expression	Comment	Ref.
XLOC_003810	25 MG patients, 25 MG-T patients, and 25 NCs	Mononuclear cells from thymus tissues/qRT-PCR	Upregulated in MG and MG-T patients’ samples.	XLOC_003810 expression led to increased number of CD4+ T cells and inflammatory cytokines; also caused decrease in PD-1/CD4+ and PD-L1/CD14+ cells percentage.	[10]
XLOC_003810	25 MG-T patients and 25 NCs	Mononuclear cells from thymus tissues/ qRT-PCR	Upregulated in thymic CD4+ T cells of MG-T patients.	XLOC_003810 expression led to Th17/Treg imbalance, which is characterized by increased expression of RORγt and Th17-related cytokines, while decreased Foxp3 as a Treg-specific transcript.	[11]
SNHG16	24 MG patients and 29 HCs	PBMC/qRT-PCR	Upregulated in PBMC samples of MG patients compared with HCs.	SNHG16 increased IL-10 expression through sponge with let-7c-5p. Furthermore, the interaction led to suppression of apoptosis and promotion of cell proliferation.	[12]
IFNG-AS1	32 MG patients and 20 HCs	PBMC/qRT-PCR	IFNG-AS1 was significantly downregulated in MG patients compared with HCs.	IFNG-AS1 downregulation was correlated with QMG and serum anti-AchR antibody positive samples. An animal study revealed its role in clinical severity reduction. It also involved in CD4+ T cells immune response through downregulation of HLA-DRB and DOB.	[13]
MALAT-1	38 MG patients and 40 HCs	PBMC/qRT-PCR	MALAT-1 was significantly downregulated in MG patients compared with HCs.	More investigation showed negative regulatory role of MALAT-1 on miR-338-3p expression. MALAT-1/miR-338-3p/MSL2 network might be potential target for MG treatment.	[14]
lncRNA & mRNA profile	48 MG patients and 26 HCs	PBMC/chip array and qRT-PCR	There was significant differences in mRNA and lncRNA expressions in MG patients compared with controls.	LncRNA XLOC 003810 and ENSG00000250850.2 were the most upregulated and downregulated transcripts, respectively. Functional annotation revealed their association with inflammatory response, NF-KB pathway, and synaptic specificity at neuromuscular junction.	[15]
lncRNA & mRNA profile	34 MG patients and 12 HCs	PBMC/chip array and qRT-PCR	There was differentially lncRNA and mRNA expression pattern between MG patients with and without thymoma.	Oebiotech_11933 lncRNA was the most upregulated transcript in patients with thymoma and associated with cellular response to interferon-γ, platelet degranulation, chemokine receptor binding, and cytokine interactions terms that are important in MG pathogenesis. It also had regulatory role in TF-lncRNA-target gene network.	[16]
lncRNA & mRNA profile	_	Human primary myoblast cells treated with recombinant human AChR IgG antibodies/RNA-seq (Next Seq 500) and qRT-PCR	AchR antibody led to deregulation of transcripts of the skeletal muscle cells.	Enrichment analysis showed related pathways such as extracellular matrix, actin cytoskeleton, myosin filament, cholesterol metabolic processes, and circadian rhythms. MEG3, RP11-184M15.1, and SNHG3 lncRNAs correlated with several mRNAs associated with the pathways.	[17]
Transcriptome profile	12 AChR + EOMG patients and 6 HCs (identification phase)/17 patients and 12 HCs (validation phase)	PBMC/RNA-seq (HiSeq 2000) and qRT-PCR	A total of 178 coding transcripts and 229 lncRNAs including pre-miRNAs were differentially expressed in MG patients.	A total of 46% of deregulated genes were associated with infectious disease and inflammatory response. miR-612, miR-3654, miR-3651, and pre-miR-3651 were upregulated in AchR-EOMG. There were no pattern differences between pre- and post-thymectomy patients.	[18]
Exosomal LncRNA profile	6 MG patients and 6 HCs (identification phase)/30 MG patients and 10 HCs (validation phase)	Serum exosome/RNA-seq (HiSeqTM 2500) and qRT-PCR	There were 378 significantly upregulated and 348 significantly downregulated exosomal lncRNAs in MG patients compared with controls.	ENST00000583253.1 was the most elevated transcript, and NR_046098.1 expression was correlated with MG grade. The 5 most significant transcripts had important role through interaction with 14 miRNA and 30 mRNA. Enrichment analysis showed their effects on immune-related pathways.	[19]

**Table 2 ijms-22-12964-t002:** Role of miRNAs in MG (PBMC: peripheral blood mononuclear cell, qRT-PCR: quantitative reverse transcription PCR, HC: healthy control, NC: negative control (cardiac surgery cases), LOMG: late-onset myasthenia gravis, EOMG: early-onset myasthenia gravis, TAMG: thymoma-associated myasthenia gravis, MGH: myasthenia gravis patients with thymus hyperplasia, OMG: ocular myasthenia gravis, GMG: generalized myasthenia gravis, RTX: rituximab, QMGS: Quantitative Myasthenia Gravis Score).

miRNA	Participants	Source of Materials and Methods	Expression	Comment	Ref.
TF–miRNA–gene network	In silico study	263 MG risk genes, 128 risk miRNAs, and 21 risk TFs through literature and databases/composite FFL motif-specific subnetwork (CFMSN)	Critical miRNAs in subnetwork were miR-20b-5p, miR-451a, miR-17-5p, miR-145-5p, miR-155-5p, miR-34a-5p, miR-20a-5p, miR-29b-5p, miR-221-5p, miR-29a-5p, let-7a-5p, let-7c-5p, and let-7 g-5p with participant of three TFs include MYC, ESR1, and BCL6. BCL2, VEGFA, KRAS, IL6, and MAPK1 were key genes in MG subnetwork.	Enrichment analysis revealed pathways related to infection and cancer. Additionally, 21 drugs were introduced through the CFMSN as a novel treatment for MG, such as estradiol, estramustine, raloxifene, and tamoxifen.	[20]
miRNA-drug pathways	In silico study	162 risk gene, 85 risk miRNA, and 45 pathways through literature and databases	miRNA-146a regulated the most pathways related to MG disease through negative regulatory role on immune-related genes include IL-8 and RANTES. Results introduced rituximab, adalimumab, sunitinib, and muromonab as potential novel drugs for MG.	Enrichment analysis revealed that the MG-related genes associated with immune disease, immune system, and cancers pathways in which hsa04060 (cytokine–cytokine receptor interaction) was the most significant pathway in MG disease.	[21]
miRNAs in ophthalmoplegic MG (OP-MG)	In silico study	miRNAs that were identified in EOMG from miRTarBase database and analysis their interactions with genes deregulated in OP-MG	miR-499 and miR-206 were highly expressed in EOMG, and they had interaction with genes related to OP-MG and strabismus pathways such as TFAM and ANK1.	IL-6 and CANX genes, which were identified from OP-MG whole exome sequencing with high expression correlations, had regulatory region for miRNA interactions.	[22]
miRNAs profile	In silico study	89 MG risk genes were identified from the literature	93 miRNAs were differentially expressed in MG patients compared with controls that mostly related to immune disease and immune system.	miR-195 had interaction with 6 MG-related pathways, which represented its global genetic regulator in MG. miRNA-mediated SNP switching pathway network showed polymorphisms role in pathway regulation, especially rs28457673, in miR-15/16/195/424/497 family effect on IGFR and cancer-related pathways.	[23]
miRNAs profile	In silico study	86 SNPs in 44 related genes and 30 miRNAs were identified from the literature	Differentially expressed miRNAs in MG patients: miR-155, miR-146a, miR-181c, let-7a and miR-15b.	miRNAs had common target genes such as MAPK1, SMAD4, SMAD2, and BCL2, which played important role in adherens, junctions, apoptosis, or cancer-related pathways, respectively. Additionally, polymorphisms in these genes could be related to disease through T cell differentiation regulation.	[24]
miRNAs and mRNAs profile	20 MG patients and 10 HCs (mRNA)/3 MG patients and 3 HCs (miRNA)	Array express database (-MEXP-518) and literature	551 upregulated and 584 downregulated mRNAs were identified in MG. Additionally, 21 and 25 miRNAs were upregulated and downregulated, respectively.	Dysregulated miRNA in MG such as miR-634 were also involved in other autoimmune disease. These miRNAs regulate key genes in MG such as MAPK1 and RAF1, which play important role in pathways crosstalk, especially immune and cancer-related ones.	[27]
miRNAs profile	Italian group: 20 responder (R) and 20 non-responder (NR) MG patients to immunosuppressive treatment	PBMC/miRnome NGS and real-time PCR on array cards	18 miRNAs were significantly upregulated in NR group versus R one, while 23 miRNAs were downregulated in NR group compared with R.	miR-323b-3p, miR-409-3p, and miR-485- 3,p which are clustered on 14q32.31 location, were significantly downregulated in NR versus R groups. While miR-181d-5p and miR-340-3p had opposite expression trend. Additionally, miR-323b-3p, miR-409-3p, and miR-485-3p had predictive value in treatment response in MG.	[25]
	Israeli group: 20 R- and 13 NR-MG patients to immunosuppressive treatment				
miRNAs profile	3 MG patients and 3 HCs (identification phase)/34 MG patients and 10 HCs (validation phase)	PBMC/Agilent Human miRNA array and Real-time PCR and Semi-Quantitative RT-PCR	16 miRNAs were upregulated, and 18 miRNAs were downregulated in MG patients.	miR-320 was downregulated in MG and led to overexpression of proinflammatory cytokines through COX-2 expression enhancement. Furthermore, miR-320 and its downstream targets regulated by ERK/ NF-κB pathways.	[26]
miRNAs profile	16 MG patients (7 germinal center (GC) positive and 9 GC negative)	Thymus blocks/microarray (GSE103812) and qRT-PCR	55 non-coding RNAs including 38 mature miRNAs showed differential expression between GC positive and negative samples.	Enrichment analysis for identified miRNAs and predicted targets revealed pathways related to humoral immune response, cellular immunity, and cytokine	[28]
				and NF kappa B signaling, which overlapped with miRNAs in SLE, RA, and autoimmune thyroid. Additional results showed miR-139-5p and miR-452-5p negatively regulate RGS13 expression, involved in GC regulation.	
miRNAs profile	12 untreated-EOMG, 8 corticoid-treated EOMG patients, and 6 NCs	Thymic biopsies/Affymetrix GeneChip miRNA 3.0 Array and RT-PCR	61 mature miRNAs and 13 pre-miRNAs were deregulated in EOMG patients compared with controls.	miR-486-5p and miR-7-5p were the most up and downregulated miRNA in both assay platforms. In vitro studies revealed miR-7-5p had a negative role on CCL21 expression, which associated with MG through thymic changes. Additionally, miR-125a regulated FOXP3 and led to downregulation of inflammatory pathway. Furthermore, miRNAs that localized near FMR1 gene were downregulated as a cluster.	[29]
miRNAs profile	3 MG patients and 3 HCs (identification phase)	PBMC/microarray, RT-PCR	21 and 23 miRNAs were significantly upregulated and down regulated in patients, respectively.	Further study confirmed let-7 family downregulation in MG patients. Additionally, IL-10 as a target gene for let-7c had negative correlation with its expression in patients and Jurkat cells.	[30]
	/34 MG patients and 10 HCs (validation phase)				
miRNAs profile	3 TAMG and 3 NCs (identification phase) / 9 TAMG patients and 9 NCs (validation phase)	Thymic biopsies/microarray and RT-PCR	104 and 33 miRNAs were significantly downregulated and upregulated in TAMG, respectively, such as let-7 family.	miR-125a-5p was significantly upregulated in patients in both phase of the study. Additionally, miR-125a-5p had negative regulatory role on FOXP3, a key gene in immune system, in patients and Jurkat cells.	[31]
miRNAs profile	4 MGH patients and 4 NCs (identification phase)/9 MGH patients and 9 NCs (validation phase)	Thymic biopsies/microarray and RT-PCR	12 and 21 miRNAs were significantly up- and downregulated in MGH patients.	Downregulation of miR-548k was verified trough RT-PCR. Further study revealed its negative regulatory role on CXCL13 expression in patients and Jurkat cells, which may relate to MGH pathogenesis.	[32]
miRNAs profile	5 patients from each group of EOMG, LOMG, TAMG, and 5 HCs (identification phase)/22 EOMG, 27 LOMG, and 12 TAMG and HCs (validation phase)	Serum/miRNA microarray and RT-PCR	32 miRNAs had differential expression levels between MG groups and healthy controls.	miR15b, miR20b, miR-192 in EOMG and miR122, miR-140-3p, miR-185, miR-375, miR885-5P in LOMG were significantly downregulated in validation phase too. Only miR15b was decreased in 3 groups of MG patients. There were no differences in miRNAs expression after immunosuppressive treatment.	[33]
miRNAs profile	EAMG rats and control rats (identification phase)/36 MG patients and 30 HCs (validation phase)	PBMC/microarray and qRT-PCR	8 downregulated and 3 upregulated miRNAs were identified in EAMG rats.	miR-145 was the top downregulated miRNAs in rats, which further validate in human samples. miR-145 negatively regulated CD28 and NFATc1, which were important in T cell clonal expansion, proliferation, and differentiation. Upregulation of miR-145 led to reduced Th17 pathogenic response.	[34]
miRNAs profile	11 MuSK + MG patients and 10 HCs	PBMC/next-generation sequencing and qRT-PCR	96 miRNAs were significantly downregulated in MuSK + MG patients compared with HC and just 5 miRNAs were upregulated in patients’ group.	miR-340-5p, miR-106b-5p, miR-27a-3p, and miR-15a-3p had most differentiation ratio between groups and also validated by qRT-PCR. They could be potential biomarkers in MuSK + MG diagnosis.	[35]
miRNAs profile	5 MuSK + MG patients and 5 HCs (identification phase)/20 MuSK + MG patients and 20 HCs (validation phase)	Serum/microRNA PCR Panel (V4.M include 179 miRNA) and qRT-PCR	Most of miRNAs in panel showed differentially expression, but only 10 miRNAs had significant differences in MG compared with HC.	Among 10 identified miRNAs, 4 miRNAs—namely, miR-151a-3p, miR-423-5p, let-7f-5p, and let-7a-5p were validated to be differentially expressed in MuSK + MG. These miRNAs had potential to be biomarkers for MuSK + MG diagnosis but not for AchR + MG.	[36]
miRNAs profile	3 female MuSK + MG patients and 3 HCs female (identification phase)/33 MuSK + MG patients and 20 HCs (validation phase)	Serum/microRNA PCR Panel and qRT-PCR	miR-210-3p and miR-324-3p were significantly decreased in MG patients, while miR-328-3p was upregulated in patients compared with controls.	Downregulation of miR-210-3p and miR-324-3p was confirmed in validation phase of study. There were no correlation between miRNAs expression and clinical features of patients.	[37]
miRNAs profile	4 AchR + MG patients and 4 HCs (identification phase)/16 AchR + MG patients and 13 HCs (validation phase)	Serum/microRNA PCR Panel I + II (V1.M include 168miRNA) and qRT-PCR	Most of miRNAs in panel showed differentially expression, but only 11 miRNAs had significant differences in MG compared with HC.	miR150-5p and miR21-5p	[38]
				were significantly elevated, while miR27a-3p was reduced in validation phase of study. Among them, miR150-5p had the strongest association with MG disease, and its reduction after thymectomy was correlated with better clinical status.	
miRNAs profile	4 AChR + generalized LOMG patients and 4 HCs (identification phase)/50 generalized LOMG and 23 ocular LOMG patients (validation phase)	Serum/microRNA human PCR Panel (V4) and qRT-PCR	miR-106b-3p, miR-30e-5p, miR-223-5p, miR-140-5p, and miR-19b-3p were significantly elevated in LOMG patients compared with HC.	miR-150-5p and miR-21-5 were lower in ocular LOMG compared with generalized patients. Additionally, miR-150-5p, miR-21-5p, and miR-30e-5p were significantly reduced after immunosuppressant treatment, and their expression was positively correlated with Myasthenia Gravis Composite (MGC) score.	[39]
miRNAs profile	5 AChR + OMG patients and 4 HC (identification phase)/83 OMG patients (validation phase)	Serum/microRNA PCR Panel (V4.M, include 179 miRNA) and qRT-PCR	30 miRNAs were significantly elevated in OMG patients compared with HC.	miR-30e-5p and miR-150-5p were validated after two-year follow-up, which were significantly higher in secondary GMGs than OMGs, which could be potential predictive biomarkers for generalization. Additionally, miR-30e-5p predictive value was near 100% for all OMGs and late onset OMG subgroup.	[40]
miRNAs profile with herbal medicine (JJN)	60 MG patients and 10 HCs	PBMC/miRNA microarray	87 miRNAs expression were significantly different between pretreatment and treated patients and healthy controls, such as let-7b-5p, miR-149-5p, let-7c, and miR-20a-5p.	QMG scores of patients 3 and 6 months after treatment with Jian Ji Ning herbal medicine were significantly lower. Its effective role could be related to inhibition of apoptotic	[41]
				pathways in immune cells and miRNAs expression regulation.	
Exosomal miRNA Profile	92 MG patients and 42 HCs	Exosomes of serum/next-generation sequencing (Illumina HiSeq 2500) and qRT-PCR	43 and 25 miRNAs were significantly down- and upregulated in MG patients compared with HC.	miR-106a-5p downregulation was validated with qRT-PCR, and its expression was lower in GMG patients compared with OMG. Additionally, its expression was more decreased in moderate–severe patients than mild MG and negatively correlated with QMGS.	[42]
miR-15b	57 MG patients (20 EOMG, 22 LOMG, and 15 TAMG) and 20 HCs	PBMC/qRT-PCR	miR-15b was significantly downregulated in all MG patients compared with HCs.	miR-15b negatively regulated IL-15, and its expression was decreased in MG patients. Further study on animal model showed overexpression on miR-15b led to downregulation of IL-15.	[43]
miR-15 cluster	32 MG patients and 20 HCs	PBMC/ RT-PCR	Expression of MiR-15 cluster (miR-15a, -15b, and 16) was significantly decreased in patients compared with controls.	miR-15a negatively regulated CXCL10 which affected T cells activation and immune response. Additionally, prednisone treatment could upregulate miR-15a in steroid-responsive patients, which led to control the disease.	[44]
miR-19b	52 TAMG, 12 thymoma patients without MG, and 11 HCs	Thymic biopsy/qRT-PCR	Expression of miR-19b-5p was significantly higher in thymoma patients with or without MG compared with HCs.	There were differences in miR-19b-5p levels between thymoma patients with and without MG. Additionally, miR-19b negatively regulated thymic stromal lymphopoietin in TAMG patients.	[45]
miR-20b	32 MG patients (18 ocular and 14 generalized) and 28 HCs	Peripheral blood/Real-time PCR	miR-20b was significantly downregulated in patients compared with controls. Additionally, expression levels were much lower in GMG than OMG.	miR-20b regulated IL-8 and IL-25 levels negatively. Corticosteroid treatments led to miR-20b upregulation and reached normal level after three month. As a result, inflammatory cytokine levels decreased.	[46]
miR-21 and miR-126	60 MG patients and 50 healthy controls	PBMC/qRT-PCR	miR-126 and miR-21 expression were significantly down- and upregulated in patients compared with controls, respectively.	Their sequences were highly conserved in human genome. These miRNAs expressions also correlated with genes related to inflammatory immune response such as AchR-AB, IL-6, and FOXP3.	[47]
miR-23b	32 MG patients and 32 HCs	Serum/qRT-PCR	miR-23b expression was significantly increased in MG patients compared with HCs.	miR-23b expression had positive correlation with QMGS in patients.	[48]
miR-27a-3p	19 MG patients (7 OMG and 12 GMG) and 17 NCs	Thymic biopsy/RT-PCR	miR-27a-3p was significantly upregulated in MG patients compared with controls.	miR-27a-3p expression level was higher in GMG than OMG, and its expression was correlated with QMG score.	[49]
miR-29 family	12 EOMG patients and 6 NCs	Thymic biopsy/RT-PCR	miR-29 family and DICER gene, which regulate INF-B, were significantly downregulated in patients compared with controls.	DICER expression was correlated with miR-29a-3p. Moreover, animal study confirmed miR-29 and DICER reduction, which led to increased INF-B and pro-inflammatory Th17 cells.	[50]
miR-133	35 MG patients and 11 HCs	Serum/qRT-PCR	miR-133 was significantly elevated in patients compared with HCs.	PAX7 was downstream target of miR-133, and regulation of it was associated with circ-FBL expression, which was upregulated in MG.	[51]
miR-146a	27 AChR-positive MG patients and 10 NCs (thymic biopsy) 31 MG patients and 11 NCs (PBMC, serum)	Thymic biopsies, PBMC, serum/RT-PCR	Expression of miR-146a was significantly higher in corticosteroid-treated patients than corticosteroid-naïve samples, while its expression in naïve group was lower than controls.	Serum results represented downregulation of miR-146a in MG patients compared with controls, which associated with TLR activation, inflammation process, and thymic hyperplastic changes. Its expression was negatively correlated with mRNA targets (IRAK1, c-REL, and ICOS). Additionally, results led to new insight in the possible mechanism of corticosteroid function in MG.	[52]
miR-146a	52 MG patients and 60 HCs	Serum/ RT-PCR	Expression of miR-146a was significantly higher in patients compared with controls.	TRAF6 was significantly upregulated as a miR-146a target in patients. ROC curve analysis revealed its potential in MG diagnosis.	[53]
miR-146a	108 MG patients and 50 HCs	PBMC/RT-PCR	Expression of miR-146a was significantly higher in patients compared with controls.	miR-146a enrichment analysis revealed its possible role in MG pathogenesis through Toll-like receptor signaling pathway, neurotransmitter regulatory signaling pathways, and EB signaling pathways.	[54]
miR-146a	20 MG patients and 21 HCs	PBMC/qRT-PCR	Expression of miR-146a was significantly higher in patients compared with controls.	Further studies in animal model showed AntagomiR-146a led to downregulation of miR-146a, CD40, CD80, TLR4, and NF-κB on AchR specific B cells.	[55]
miR-150-5p and miR-146a-5p	12 AChR + refractory MG patients before and 6 months after treatment with low-dose RTX	Serum exosomes/RT-PCR	RTX led to significant reduction in miR-150-5p expression.	miR-150-5p reduction caused decrease in clinical scores and prednisolone requirement. Additionally, miR-150-5p had positive correlation with CD19+ and CD27+ B cells count.	[56]
miR150-5p	30 GMG patients and 30 HCs	Serum/qRT-PCR	miR150-5p was significantly upregulated in patients compared with controls.	Cytokine profile revealed differentially serum levels between patients and controls. Furthermore miR-150-5p serum levels were positively and negatively correlated with IL-10 and IL-17 levels, respectively.	[57]
miR-150	40 AChR + EOMG patients and 19 NCs (thymic biopsy)/43 patients (27 = PBMC and 16 = serum) And HCs (14 = PBMC & 11 = serum)	Thymic biopsies, PBMC, serum/RT-PCR	MiR-150 expression level was significantly higher in EOMG thymus compared with controls. It was also more elevated in patients with high degree of thymic hyperplasia than lower ones.	Peripheral blood study revealed miR-150 downregulation in patients, especially in 4+ 15 T cells. Furthermore, anti-miR-150 treatment led to overexpression of pro-apoptotic genes such as P53 and AIFM2, also regulated MYB expression, which suggests its regulatory role on immune cells.	[58]
miR-150-5p and miR-21-5p	80 AChR + MG patients undergo thymectomy with prednisone treatment and prednisone treatment alone	Serum (at 12, 24, and 36 months after baseline)/qRT-PCR	miR-150-5p was significantly reduced at 24 months after thymectomy, while miR-21-5p was elevated at 36 months after thymectomy.	The results represented miR-150-5p as a disease-specific biomarker in AChR + MG. Furthermore, miR-21-5p expression level was negatively correlated with prednisone dose in MG patients.	[59]
miR-150-5p and miR-21-5p	71 MG patients 23 other autoimmune disorder and 55 HCs	Serum/qRT-PCR	miR-150-5p and miR-21-5p were elevated in MG patients before treatment compared with healthy controls.	6 months after immunosuppressive treatments both miRNAs levels were significantly lower than before treatment evaluation.	[60]
miR-150-5p and miR-21-5p	10 MG patients with mild disease before and after 12 weeks physical performance	Serum/RT-qPCR	miR-150-5p and miR-21-5p were significantly decreased after physical exercise course.	Physical exercise could be safe for well regulated-MG patients.	[61]
let-7a-5p,	12 Musk + MG patients before and 6 months after low dose RTX	Serum/qRT-PCR	miR-151a-3p was significantly reduced (28.1%) after RTX treatment.	Its expression correlated with clinical severity and reduced prednisone requirement after RTX treatment.	[62]
let-7f-5p, miR-151a-3p, and miR-423-5p					
miR-155	MG patients and healthy controls	PBMC/qRT-PCR	miR-155 was significantly upregulated in MG patients compared with controls.	Dexamethasone treatment caused symptom improvement through decreased miR-155 expression, which led to disturbance in the antibody class switching.	[63]
miR-155	32 MG patients and 31 HC	PBMC/qRT-PCR	miR-155 was significantly upregulated in MG patients compared with controls.	miR-155 silencing impaired the BAFF-R/NF-κB signaling pathways and reduced AChR-specific autoantibodies.	[64]
miR-181c	22 AchR + MG patients (12 OMG and 10 GMG) and 20 HCs	PBMC/RT-PCR	miR-181c was significantly downregulated in MG patients, especially lower in GMG group.	miR-181c negatively regulates IL-7, also IL-7 effect of Th17 cytokine production. miR-181c overexpression led to decrease both inflammation cytokine level, IL7, and IL-17.	[65]
miR-522-3p	30 TAMG, 20 MG patients without thymoma and 15 HCs	Thymic biopsy and peripheral blood/RT-PCR	miR-522-3p was significantly downregulated in MG patients compared with HCs. Additionally, its expression was lower in TAMG compared with MG without thymoma.	miR-522-3p negatively regulated SLC31A1 expression. SLC31A1 overexpression had positive effects on cell viability, cycle progression, and the levels of IL-2 and IL-10 in Jurkat cells.	[66]

## Data Availability

Not applicable.

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
