# Peer review of "A Review on the Role of Non-Coding RNAs in the Pathogenesis of Myasthenia Gravis"

_ijms, 2021, doi:10.3390/ijms222312964_

Round 1
Reviewer 1 Report
The topic is well presented. However there is the feeling that the article is a list, without exploring some of the information and implications that are cited.
Introduction could be improved to present better the characteristics of the disease.
Discussion is fine, but a bit generic. Maybe a more deep description or analysis of just one of the topic presented in the article could significantly improve it.
There are some minor corrections to do:
96) Some typos in the legend of Table 1.
115) typos in orbicularis. Moreover the whole sentence is not clear.
132) Conclusions: …
151) Some typos in the legend of Table 2. There are typos in Table 2 also (insilco, jurket, ecc…). Finally, Table 2 should have some sort of alphabetic order to make more easy the reading when searching for a specific miRNA or cluster.
175) QMG should be briefly explained in the article.
Finally:
Reference [26] is missing in the text.
Reference [32] doesn't seems the correct one.
Is miR-145, not 146, the one described in reference [33].
Is miR-126, not 26, the one described in reference [52].
Author Response
- We added some notes in the Introduction.
- We improved discussion.
- We corrected the typos.
- We ordered miRNAs in Table 2.
- We corrected the mentioned references.
Reviewer 2 Report
This review sheds some light on the possible implication of non-coding RNAs in the development of Myastenia Gravis.
The theme of this research is interesting; however, I have some minor suggestions to make:
- I would include a more extensive future implication of this matter in clinical medicine. The review is otherwise incomplete.
-A proper conclusion section is missing.
Author Response
We added future perspectives and conclusions.